# CRYSFORMER: PROTEIN STRUCTURE PREDICTION VIA 3D PATTERSON MAPS AND PARTIAL STRUCTURE ATTENTION

## ABSTRACT

Determining the structure of a protein has been a decades-long open question. A protein's three-dimensional structure often poses nontrivial computation costs, when classical simulation algorithms are utilized. Advances in the transformer neural network architecture –such as AlphaFold2– achieve significant improvements for this problem, by learning from a large dataset of sequence information and corresponding protein structures. Yet, such methods only focus on sequence information; other available prior knowledge, such as protein crystallography and partial structure of amino acids, could be potentially utilized. To the best of our knowledge, we propose the first transformer-based model that directly utilizes protein crystallography and partial structure information to predict the electron density maps of proteins. Via two new datasets of peptide fragments (2-residue and 15-residue) , we demonstrate our method, dubbed `CrysFormer`, can achieve accurate predictions, based on a much smaller dataset size and with reduced computation costs.

## 1 INTRODUCTION

Proteins, the biological molecular machines, play a central role in the majority of cellular processes (Tanford & Reynolds, 2004). The investigation of a protein's structure is a classic challenge in biology, given that its function is dictated by its specific conformation. Proteins comprise long chains of linked, relatively small organic molecules called *amino acids*, with a set of twenty of them considered as standard. However, these underlying polypeptide chains fold into complex three-dimensional structures, as well as into larger assemblies thereof. Consequently, biologists aim to establish a standardized approach for experimentally determining and visualizing the overall structure of a protein at a low cost.

In the past decades, there have been three general approaches to the protein structure problem: $i$) ones that rely on physical experimental measurements, such as X-ray crystallography, NMR, or cryo-electron microscopy; see (Drenth, 2007) for more details; $ii$) protein folding simulation tools based on thermodynamic or kinetic simulation of protein physics (Brini et al., 2020; Sippl, 1990); and, $iii$) evolutionary programs based on bioinformatics analysis of the evolutionary history of proteins (Šali & Blundell, 1993; Roy et al., 2010).

Recent advances in machine learning (ML) algorithms have inspired a fourth direction which is to train a deep neural network model on a combination of a large-scale protein structure data set (i.e., the Protein Data Bank (wwPDB consortium, 2019)) and knowledge of the amino acid sequences of a vast number of homologous proteins, to directly predict the protein structure from the protein's amino acid sequence. Recent research projects –such as Alphafold2 (Jumper et al., 2021)– further show that, with co-evolutionary bioinformatic information (e.g., multiple sequence alignments), deep learning can achieve highly accurate predictions in most cases.

**Our hypothesis and contributions.** While it is true that computational methods of predicting structures without experimentally confirming data are improving, they are not yet complete –in terms of the types of structures that can be predicted– and suffer from lack of accuracy in many of the details (Terwilliger et al., 2023). X-ray crystallographic data continues to be a gold standard for critical details describing chemical interactions of proteins.

Having a robust and accurate way of going directly from an X-ray diffraction pattern to a solved structure would be a strong contribution to the field of X-ray crystallography. Such approaches are missing from the literature, with the exception of Pan et al. (2023), a recent effort on the same problem based on residual convolutional autoencoders.

Here, we present the first transformer-based model that utilizes protein crystallography and partial structure information to directly predict the electron density maps of proteins, going one step beyond such recent approaches. While not yet ready to solve real problems, we demonstrate success on a simplified problem. As a highlight, using a new dataset of small peptide fragments of variable unit cell sizes –a byproduct of this work– we demonstrate that our method, named `CrysFormer`, can achieve more accurate predictions than state of the art (Pan et al., 2023) with less computations.

Some of our findings and contributions are:

- `CrysFormer` is able to process the global information in Patterson maps to infer electron density maps; to the best of our knowledge, along with Pan et al. (2023), these are the first works to attempt this setting.
- `CrysFormer` can incorporate "partial structure" information, when available; we also show that such information could be incorporated in existing solutions that neglected this feature, like the convolutional `U-Net`-based architectures in Pan et al. (2023). However, the `CrysFormer` architecture still leads to better reconstructions.
- In practice, `CrysFormer` achieves a significant improvement in prediction accuracy in terms of both Pearson coefficient and mean phase error, while requiring both a smaller number of epochs to converge and less time taken per epoch.
- This work introduces a new dataset of variable-cell dipeptide fragments, where all of the input Patterson and output electron density maps were derived from the Protein Databank (PDB) (wwPDB consortium, 2019), solved by X-ray Crystallography. We will make this dataset publicly available.

## 2 PROBLEM SETUP AND RELATED WORK

**X-ray crystallography and the crystallographic phase problem.** X-ray crystallography has been the most commonly used method to determine a protein's electron density map[1] for over 100 years (Lattman & Loll, 2008). However, there is an open question, called the crystallographic phase problem, that prevents researchers from utilizing it to predict true structures/electron density maps.

In review, each spot (known as a reflection) in an X-ray crystallography diffraction pattern is denoted by three indices $h, k, l$, known as Miller indices (Ashcroft & Mermin, 2022). These correspond to sets of parallel planes within the protein crystal's unit cell that contribute to producing the reflections. The set of possible $h, k, l$ values is determined by the radial extent of the observed diffraction pattern. Any reflection has an underlying mathematical representation, known as a structure factor, dependent on the locations and scattering factors of all the atoms within the crystal's unit cell. In math:

$$F(h, k, l) = \sum_{j=1}^{n} f_j \cdot e^{2\pi i (h x_j + k y_j + l z_j)}, \tag{1}$$

where the scattering factor and location of atom $j$ are $f_j$ and $(x_j, y_j, z_j)$, respectively.

A structure factor $F(h, k, l)$ has both an amplitude and a phase component (denoted by $\phi$) and thus can be considered a complex number. Furthermore, suppose we knew both components of the structure factors corresponding to all of the reflections within a crystal's diffraction pattern. Then, in order to produce an accurate estimate of the electron density at any point $(x, y, z)$ within the crystal's unit cell, we would only need to take a Fourier transform of all of these structures, as in:

$$\rho(x, y, z) = \frac{1}{V} \cdot \sum_{h,k,l} |F(h, k, l)| \cdot e^{-2\pi i (h x + k y + l z - \phi(h,k,l))}, \tag{2}$$

where $V$ is the volume of the unit cell. The amplitude $|F(h, k, l)|$ of any structure factor is easy to determine, as it is simply proportional to the square root of the measured intensity of the corresponding reflection. However, it is impossible to directly determine the phase $\phi(h, k, l)$ of a structure factor, and this is what is well-known as the crystallographic phase problem (Lattman & Loll, 2008).

---

[1]The electron density is a measure of the probability of an electron being present just around a particular point in space; a complete electron density map can be used to obtain a molecular model of the unit cell.

**Solving the phase problem.** Various methods have been developed to solve the crystallography phase problem. The three commonly used methods are isomorphous replacement, anomalous scattering, and molecular replacement (Lattman & Loll, 2008; Jin et al., 2020). Also, what is known as direct methods have been successful for small molecules that diffract to atomic resolution, but they rarely work for protein crystallography, due to the difficulty of resolving atoms as separate objects. Alternative methods have been developed to solve the phase problem based on intensity measurements alone, known as phase retrieval (Guo et al., 2021; Kappeler et al., 2017; Rivenson et al., 2018). However, these methods have not been widely used in X-ray crystallography, because they assume different sampling conditions or were designed for non-crystallographic fields of physics. The iterative non-convex Gerchberg–Saxton algorithm (Fienup, 1982; Zalevsky et al., 1996) is a well-known example of such methods, but requires more measurements than is available in crystallography.

Although adaptations of the Gerchberg–Saxton algorithm have been proposed for crystallography-like settings, they have not been used to solve the phase problem except in special cases where crystals have very high solvent content (He & Su, 2015; He et al., 2016; Kingston & Millane, 2022). More recently, Candes et al. (2013) introduced the `Phaselift` method, a convex, complex semidefinite programming approach, and Candes et al. (2015) the Wirtinger flow algorithm (Candes et al., 2015), a non-convex phase retrieval method; both these methods have not been applied practically, due to their computationally intensive nature.

## 3 CRYSFORMER: USING 3D MAPS AND PARTIAL STRUCTURE ATTENTION

Inspired by (Hurwitz, 2020), we rely on deep learning solutions to directly predict the electron density map of a protein. Later in the text, we demonstrate that such a data-centric method achieves both better accuracy and reduced computational cost.

**The Patterson function.** We utilize the *Patterson function* (Patterson, 1934), a simplified variation of the Fourier transform from structure factors to electron density, in which all structure factor amplitudes are squared, and all phases are set to zero (i.e., ignored), as in:

$$p(u, v, w) = \frac{1}{V} \cdot \sum_{h,k,l} |F(h, k, l)|^2 \cdot e^{-2\pi i (hu + kv + lw)}. \tag{3}$$

It is important to note the Patterson map can be directly obtained from raw diffraction data without the need for additional experiments, or any other information.

Due to the discrete size of the input and output layers in deep learning models, we can discretize and reformulate the electron density map –and its corresponding Patterson map– as follows: Suppose the electron density map of a molecule in interest is discretized into a $N_1 \times N_2 \times N_3$ 3d grid. The electron density map can then be denoted as $\mathbf{e} \in \mathbb{R}^{N_1 \times N_2 \times N_3}$. The Patterson map is then formulated as follows, where $\odot$ means matrix element-wise multiplication:

$$\mathbf{p} = \Re \left( \mathcal{F}^{-1} \left( \mathcal{F}(\mathbf{e}) \odot \mathcal{F}(\widehat{\mathbf{e}}) \right) \right) \approx \Re \left( \mathcal{F}^{-1} \left( |\mathcal{F}(\mathbf{e})|^2 \right) \right).$$

Breaking down the above expression, $\mathcal{F}(\mathbf{e}) \odot \mathcal{F}(\widehat{\mathbf{e}}) \approx |\mathcal{F}(\mathbf{e})|^2$ denotes only the magnitude part of the complex signals, as measured through the Fourier transform of the input signal $\mathbf{e}$. Here, $\widehat{\mathbf{e}}$ denotes an inverse-shifted version of $\mathbf{e}$, where its entries follow the shifted rule as in $\widehat{e}_{i,j,k} = e_{N-i,N-j,N-k}$.

**Using deep learning.** We follow a data-centric approach and train a deep learning model, abstractly represented by $g(\boldsymbol{\theta}, \cdot)$, such that given a Patterson map $\mathbf{p}$ as input, it generates an estimate of an electron density map, that resembles closely the true map $\mathbf{e}$. Formally, given a data distribution $\mathcal{D}$ and $\{\mathbf{p}_i, \mathbf{e}_i\}_{i=1}^n \sim \mathcal{D}$, where $\mathbf{p}_i \in \mathbb{R}^{N_1 \times N_2 \times N_3}$ is the Patterson map that corresponds to the true data electron density map, $\mathbf{e}_i \in \mathbb{R}^{N_1 \times N_2 \times N_3}$, deep learning training aims in finding $\boldsymbol{\theta}^\star$ as in:

$$\boldsymbol{\theta}^\star = \arg \min_{\boldsymbol{\theta}} \left\{ \mathcal{L}(\boldsymbol{\theta}) := \frac{1}{n} \sum_{i=1}^n \ell(\boldsymbol{\theta}; \, g, \{\mathbf{p}_i, \mathbf{e}_i\}) = \frac{1}{n} \sum_{i=1}^n \|g(\boldsymbol{\theta}, \mathbf{p}_i) - \mathbf{e}_i\|_2^2 \right\}.$$

Since we have a regression problem, we use mean squared error as the loss function $\mathcal{L}(\boldsymbol{\theta})$.

**Using partial protein structures.** Due to the well-studied structure of amino acids, we aim to optionally utilize standardized *partial structures* to aid prediction, when they are available. For example, let $\mathbf{u}_i^j \in \mathbb{R}^{N_1 \times N_2 \times N_3}$ be the known standalone electron density map of the $j$-th amino acid

of the $i$-th protein sample, in a standardized conformation. Abstractly, we then aim to optimize:

$$\boldsymbol{\theta}^{\star} = \arg\min_{\boldsymbol{\theta}} \left\{ \mathcal{L}(\boldsymbol{\theta}) := \frac{1}{n} \sum_{i=1}^{n} \ell(\boldsymbol{\theta};\, g, \{\mathbf{p}_i, \mathbf{e}_i, \mathbf{u}_i^j\}) = \frac{1}{n} \sum_{i=1}^{n} \|g(\boldsymbol{\theta}, \mathbf{p}_i, \mathbf{u}_i^j) - \mathbf{e}_i\|_2^2 \right\}.$$

**Challenges and Design Principles.** We face the difficult learning problem to infer electron density maps $\mathbf{e}$ from Patterson maps $\mathbf{p}$, which involves Fourier transformations. *These transformations can be intuitively considered as transforming local information to global information*, which is rare in common deep model use cases. Secondly, it is nontrivial to incorporate the partial structure density maps $\mathbf{u}_i^j$ to aid prediction. Thirdly, the 3d data format of both our inputs and outputs often increases substantially the computational requirements. Finally, since part of our contributions is novel datasets on this problem, we need to be data efficient due to the expensive dataset creation cost. Thus, the main design principles for our model can be summarized as:

- *Design Principle #1*: Be able to process the global information in Patterson maps to correctly infer the corresponding electron density maps;
- *Design Principle #2*: Be able to incorporate partial structure information, when available;
- *Design Principle #3*: Learn to fulfill the above, with reduced computational and data-creation costs.

**Gap in current knowledge.** As an initial attempt, the well-established convolution-based `U-Net` model (Ronneberger et al., 2015) could be utilized for this task. This is the path followed in (Pan et al., 2023). However, classical `U-Net`s cannot fulfill the design principles above, since: $i$) they mostly rely on local information within CNN layers; such a setup is not suitable when Patterson maps are available, since the latter do not have meaningful local structures. $ii$) It is not clear (or, at best, non-trivial) to incorporate any partial protein structures prior information, since the latter is in a different representation domain, compared to Patterson maps. Finally, $iii$) a large 3d `U-Net` model is computationally expensive and inefficient, due to the 3d filter convolution computation.

**Our proposal: `CrysFormer`.** We propose `CrysFormer`, a novel, 3d Transformer model (Vaswani et al., 2017; Chen et al., 2021) with a new self-attention mechanism to process Patterson maps and partial protein structures, to directly infer electron density maps with reduced costs.

Inspired by recent research on the potential connection between Fourier transforms and the self-attention mechanism, found in the Transformer model (Lee-Thorp et al., 2022), `CrysFormer` captures the global information in Patterson maps and "translates" it into correct electron density map predictions, via our proposed self-attention mechanism (*Design Principle #1*). `CrysFormer` does not need an encoder-decoder structure (Vaswani et al., 2017) and artificial information bottlenecks (Cheng et al., 2019) –as in the `U-Net` architecture– to force the learning of global information.

By definition, `CrysFormer` is able to handle additional partial structure information, which comes from a different domain than the Patterson maps (*Design Principle #2*; more details below).

Finally, by using efficient self-attention between 3d image patches, we can significantly reduce the overall computation cost. Detaching our model from an encoder-decoder architecture further reduces the required depth of the model and, thus, the overall training cost (*Design Principle #3*).

**The architecture of the `CrysFormer`.** We follow ideas of a 3d visual Transformer (Chen et al., 2021) by partitioning the whole input 3d Patterson map $\mathbf{p}_i \in \mathbb{R}^{N_1 \times N_2 \times N_3}$ input into a set of smaller 3d patches. We embed them into one-dimensional "word tokens", and feed them into a multi-layer, encoder-only Transformer module. If partial structures $\mathbf{u}_i^j$ are also available, we will partition them into 3d patches and embed them into additional tokens that are sent to each self-attention layer. This way, the tokens in each layer can also "attend" the election density of partial structures, as a reference for final global electron density map predictions. Finally, we utilize a 3d convolutional layer to transform "word-tokens" back into a 3d electron density map.[2] See Figure 1.

Mathematically, we report the following: The first part is the preprocessing and partitioning of input Patterson maps $\mathbf{p}$ and additional partial structures $\mathbf{u}^j$ into 3d patches of size $d_1 \times d_2 \times d_3$. We embed those patches into one-dimensional tokens with dimension $d_t$, using of a small MLP, and add them with a learned positional embedding; this holds for both Patterson maps and structures, as below:

---

[2]We also utilize 3d convolutional layer(s) at the very beginning of the execution to expand the number of channels of the Patterson map (and potentially partial structure) inputs.

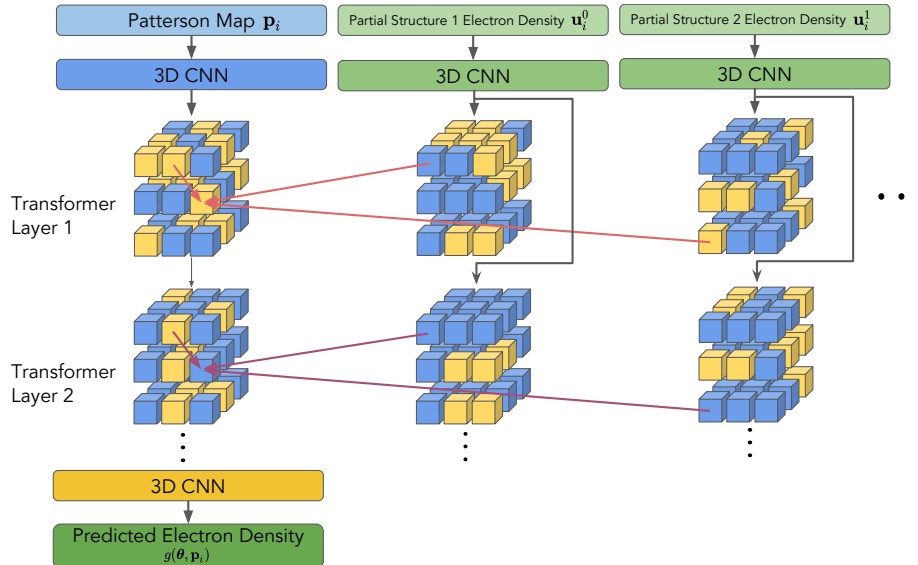

Figure 1: Abstract depiction of the `Crysformer`, which utilizes a one-way attention mechanism (red and purple arrows) to incorporate the partial structure information. The tokens from the additional partial structure all come from initial 3d CNN embedding and are not passed to the next layer.

**Patterson maps** $\mathbf{p}$          **Partial structures** $\mathbf{u}^j$

$$\mathbf{X}^0 = \mathtt{3DCNN}_{\mathbf{W}_c}(\mathbf{p}) \in \mathbb{R}^{c \times N_1 \times N_2 \times N_3}$$

$$\mathbf{X}^0 = \mathtt{Partition}(\mathbf{X}^0) \in \mathbb{R}^{\frac{N_1}{d_1} \times \frac{N_2}{d_2} \times \frac{N_3}{d_3} \times (cd_1d_2d_3)}$$

$$\mathbf{X}^0 = \mathtt{Flatten}(\mathbf{X}^0) \in \mathbb{R}^{\frac{N_1N_2N_3}{d_1d_2d_3} \times (cd_1d_2d_3)}$$

$$\mathbf{X}^0 = \mathtt{MLP}_{\mathbf{W}_c}(\mathbf{X}^0) \in \mathbb{R}^{\frac{N_1N_2N_3}{d_1d_2d_3} \times d_t}$$

$$\mathbf{X}^0 = \mathbf{X}^0 + \mathtt{PosEmbedding}(\tfrac{N_1N_2N_3}{d_1d_2d_3})$$

$$\mathbf{U}^j = \mathtt{3DCNN}_{\mathbf{W}_p}(\mathbf{u}^j) \in \mathbb{R}^{c \times N_1 \times N_2 \times N_3}$$

$$\mathbf{U}^j = \mathtt{Partition}(\mathbf{U}^j) \in \mathbb{R}^{\frac{N_1}{d_1} \times \frac{N_2}{d_2} \times \frac{N_3}{d_3} \times (cd_1d_2d_3)}$$

$$\mathbf{U}^j = \mathtt{Flatten}(\mathbf{U}^j) \in \mathbb{R}^{\frac{N_1N_2N_3}{d_1d_2d_3} \times (cd_1d_2d_3)}$$

$$\mathbf{U}^j = \mathtt{MLP}_{\mathbf{W}_p}(\mathbf{U}^j) \in \mathbb{R}^{\frac{N_1N_2N_3}{d_1d_2d_3} \times d_t}$$

$$\mathbf{U}^j = \mathbf{U}^j + \mathtt{PosEmbedding}(\tfrac{N_1N_2N_3}{d_1d_2d_3})$$

As shown in Figure 1, we design an efficient attention mechanism such that $i$) only tokens from Patterson maps attend tokens from the partial structures; $ii$) the tokens from the additional partial structures are not passed to the next layer. This is based on that the partial structure electron density information should be used by the model as a stable reference to attend to in each layer.

This one-way attention also greatly reduces the overall communication cost. In particular, let the token sequence length be $S = \frac{N_1N_2N_3}{d_1d_2d_3}$ and let $d_h$ denote the dimension of the attention head. Assuming we have $H$ attention heads and $L$ layers, `CrysFormer` uses the following attention mechanism:

$$\mathbf{U} = \mathtt{Concat}_{j=1}^{J}(\mathbf{U}^j) \in \mathbb{R}^{(SJ) \times d_t}$$

$$\mathbf{A}^h = \mathtt{Softmax}\left((\mathbf{W}_q^h\mathbf{X}^\ell)^\top(\mathtt{Concat}(\mathbf{W}_k^h\mathbf{X}^\ell, \mathbf{W}_{k'}^h\mathbf{U})\right) \in \mathbb{R}^{S \times (S+SJ)};$$

$$\widehat{\mathbf{V}}^h = \mathbf{A}^h\left(\mathtt{Concat}(\mathbf{W}_v^h\mathbf{X}^\ell, \mathbf{W}_{v'}^h\mathbf{U})\right) \in \mathbb{R}^{S \times d_h};$$

$$\mathbf{O} = \mathbf{W}_o\mathtt{Concat}\left(\widehat{\mathbf{V}}^0, \widehat{\mathbf{V}}^1, \ldots, \widehat{\mathbf{V}}^H\right) \in \mathbb{R}^{S \times d_t};$$

$$\mathbf{X}^{\ell+1} = \mathbf{W}_{\mathrm{ff2}}(\mathtt{ReLU}(\mathbf{W}_{\mathrm{ff1}}\mathbf{O})),$$

where, omitting the layer index, $\mathbf{W}_q^h$, $\mathbf{W}_k^h$, $\mathbf{W}_v^h$ are the trainable query, key, and value projection matrices of the $h$-th attention head for tokens from the Patterson map, and $\mathbf{W}_{k'}^h$, $\mathbf{W}_{v'}^h$ are the corresponding matrices for tokens from the partial structure, each with dimension $d_h$. Further, $\mathbf{W}_{\mathrm{ff1}}$ and $\mathbf{W}_{\mathrm{ff2}}$ are the trainable parameters of the fully-connected layers. We omit skip connections and layer normalization modules just to simplify notation, but these are included in practice.

As a final step, we transform the output embedding back to a 3d electron density map, as follows:

$$g(\boldsymbol{\theta}, \mathbf{p}) = \tanh(\mathtt{3DCNN}_{\mathbf{W}_o}(\mathtt{Rearrange}(\mathtt{MLP}(\mathbf{X}^L)))) \in \mathbb{R}^{N_1 \times N_2 \times N_3},$$

and, as stated previously, we use as our loss function the standard mean squared error loss.

## 4 NEW DATASETS

We generate datasets of protein fragments, where input Patterson and output electron density maps are derived from Protein Databank (PDB) entries of proteins solved by X-ray Crystallography (wwPDB consortium, 2019). We start from a curated basis of $\sim 24,000$ such protein structures. Then from a random subset of about half of these structures, we randomly select and store segments of adjacent amino acid residues. These examples are consisted of dipeptides (two residues) and 15-residues, leading to two datasets that we introduce with this work. The latter dataset contains 15 residues, where at most 3 residues could be shared between different examples. Using the `pdbfixer` Python API (Eastman et al., 2017), we remove all examples that either contain nonstandard residues or have missing atoms from our initial set. We also apply a few standardized modifications.

For our dipeptide dataset, we then iteratively expand the unit cell dimensions for each example, starting from the raw $\max - \min$ ranges in each of the three axis directions, attempting to create a minimal-size unit cell where the minimum atomic contact is at least 2.75 Angstroms (Å).[3] For our 15-residue dataset, we instead place atoms in fixed unit cells of size 41 Å x 30 Å x 24 Å to simplify the now much harder problem. After this, all examples that still contain atomic contacts of less than 2.75 Å are discarded. The examples are then reoriented via a reindexing operation, such that the first axis is always the longest and the third axis is always the shortest.

One issue leading to potential ambiguity in interpreting Patterson maps is their invariance to translation of the entire corresponding electron density (Hurwitz, 2020). To tackle this, we center all atomic coordinates such that the center of mass is in the center of the corresponding unit cell. This means that our model's predicted electron densities would always be more or less centered in the unit cell. We note that this is also the case for the majority of actual protein crystals.

Structure factors for each remaining example, as well as those for the corresponding partial structures for each of the present amino acids, are generated using the `gemmi sfcalc` program (Wojdyr, 2022) to a resolution of 1.5 Å. An electron density and Patterson map for each example are then obtained from those structure factors with the `fft` program of the `CCP4` program suite (Read & Schierbeek, 1988; Winn et al., 2011); partial structure densities are obtained in the same manner. We specify a grid oversampling factor of 3.0, resulting in a 0.5 Å grid spacing in the produced maps. All these maps are then converted into PyTorch tensors. We then normalize the values in each of the tensors to be in the range $[-1, 1]$. Since, in our PyTorch implementation, all examples within a training batch are of the same size, we remove all examples from the tensor-size bins containing fewer examples than a specified minimum batch size.

## 5 EXPERIMENTS

**Baselines.** There are no readily available off-the-self solutions for our setting, as our work is one of the first of this kind. As our baseline, we use a CNN-based `U-Net` model (Pan et al., 2023); this architecture is widely used in image transformation tasks (Ronneberger et al., 2015; Yan et al., 2021).

For comparison, we have further enhanced this vanilla `U-Net` with $i$) additional input channels to incorporate the partial structure information, despite being evidently unsound; and $ii$) a refining model procedure, which retrains the `U-Net` using previous model predictions as additional input channels. Both of these extensions are shown to greatly improve the performance of the vanilla `U-Net`. We refer the reader to the appendix for more details on our baseline model architecture.

**Metrics.** During testing, we calculate the Pearson correlation coefficient between the ground truth targets $\mathbf{e}$ and model predictions $g(\boldsymbol{\theta}, \mathbf{p})$; the larger this coefficient is, the better. Let us denote a model prediction as $\mathbf{e}'$. We define $\bar{\mathbf{e}} = \frac{1}{N_1 N_2 N_3} \sum_{i,j,k} \mathbf{e}_{i,j,k}$ and $\bar{\mathbf{e}}' = \frac{1}{N_1 N_2 N_3} \sum_{i,j,k} \mathbf{e}'_{i,j,k}$. Then, the Pearson correlation coefficient between $\mathbf{e}$ and $\mathbf{e}'$ is as below:

$$\mathtt{PC}(\mathbf{e}, \mathbf{e}') = \frac{\sum_{i,j,k=1}^{N_1,N_2,N_3}(\mathbf{e}'_{i,j,k} - \bar{\mathbf{e}}')(\mathbf{e}_{i,j,k} - \bar{\mathbf{e}})}{\sqrt{\sum_{i,j,k=1}^{N_1,N_2,N_3}(\mathbf{e}'_{i,j,k} - \bar{\mathbf{e}}') + \epsilon} \cdot \sqrt{\sum_{i,j,k=1}^{N_1,N_2,N_3}(\mathbf{e}_{i,j,k} - \bar{\mathbf{e}}) + \epsilon}}, \qquad (4)$$

---

[3]An Angstrom is a metric unit of length equal to $10^{-10}$m.

where $\epsilon$ is a small constant to prevent division by zero. To demonstrate how well our methods solve the phase problem, we also perform phase error analysis on our models' final post-training predictions using the `cphasematch` program of the `CCP4` program suite (Cowtan, 2011). We report the mean phase errors of our predictions in degrees, as reported by `cphasematch`, where a smaller phase error is desirable. Finally, we compare the convergence speed and computation cost of both methods.

| Method | Mean $\mathrm{PC}(\mathbf{e}, \mathbf{e}')$ | Mean Phase Error | Epochs | Time per epoch (mins.) |
|---|---|---|---|---|
| `U-Net` (Pan et al., 2023) | 0.735 | 67.40° | 50 | 28.93 |
| `U-Net+R` (This work) | 0.775 | 58.67° | 90 | 29.06 |
| `U-Net+PS+R` (This work) | 0.839 | 51.34° | 90 | 29.31 |
| `CrysFormer` (This work) | **0.939** | **35.16°** | **35** | **12.37** |

Table 1: `CrysFormer` versus baselines on the dipeptide dataset. `U-Net+R` refers to adding the refining procedure to `U-Net` training; `U-Net+PS+R` refers to adding further partial structures as additional channels.

**Results on two-residues.** A summary of our results on our dipeptide dataset, which consisted of $1,894,984$ training and $210,487$ test cases, is provided in Table 1. Overall, `CrysFormer` achieves a significant improvement in prediction accuracy in terms of both the Pearson coefficient and phase error, while requiring a shorter time (in epochs) to converge. `CrysFormer` also incurs much less computation cost which results in significantly reduced wall clock time per epoch.

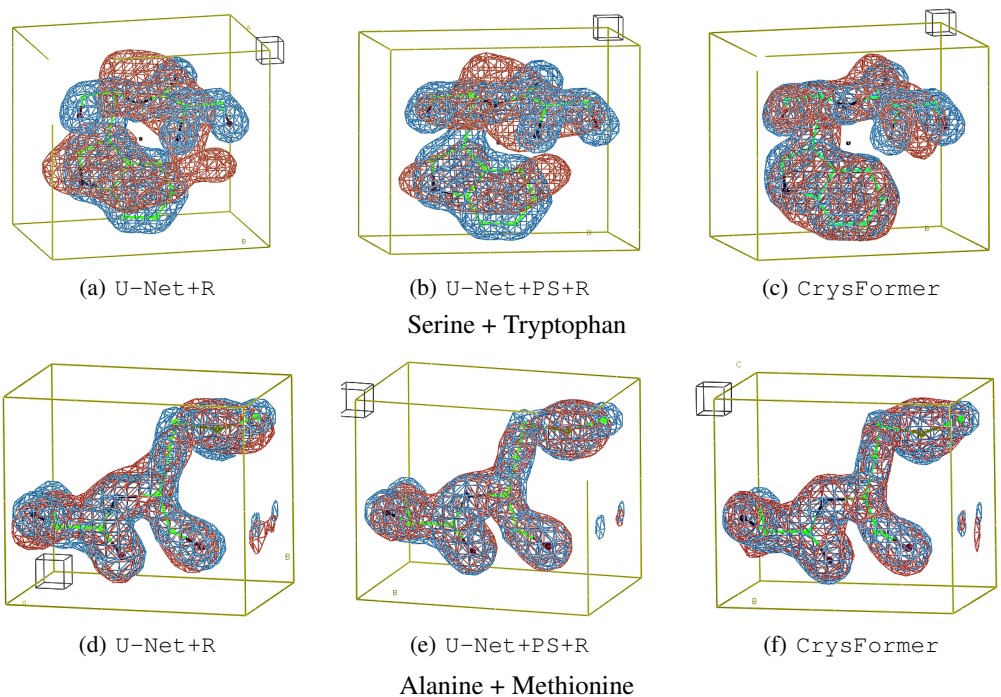

(a) `U-Net+R`        (b) `U-Net+PS+R`        (c) `CrysFormer`

Serine + Tryptophan

(d) `U-Net+R`        (e) `U-Net+PS+R`        (f) `CrysFormer`

Alanine + Methionine

Figure 2: Visualization of electron density predictions for baselines and `CrysFormer`: Ground truth density maps are shown in blue, while predictions are shown in red. The model used to generate the ground truth electron density is shown in stick representation for reference.

We further visualize some of the predictions in Figure 2, comparing side by side those made by the baselines and the `CrysFormer`. `CrysFormer` produces more accurate predictions in terms of both global and local structures. This verifies our hypothesis that $i$) the self-attention mechanism can better capture the global information in Patterson maps, and $ii$) the removal of the `U-Net`'s encoder-decoder structure prevents loss of information and improves the reproduction of finer details.

E.g., the top row of Figure 2 represents a class of examples containing a large aromatic residue, Tryptophan. `U-Net+R` models consistently produce poor predictions in this case, while the `CrysFormer` better handles such residues. `U-Net+PS+R` shows that both providing additional input channels and using the refining procedure improves results even for `U-Net` architectures; yet, `CrysFormer` still provides better reconstruction. More visualizations can be found in the appendix.

We further plot the calculated average mean phase errors of the predictions of our models against reflection resolution, see left panel of Figure 3. The predictions made by `CrysFormer` have lower mean phase error, compared to baselines. This means that the `CrysFormer` predictions, on average, can reproduce better the general shape, as well as finer details of the ground truth electron densities.

Finally, we generate a chart of the fraction of our models' predictions for which the calculated mean phase error is $< 60°$ at various ranges of resolution. We consider such predictions to accurately reproduce the level of detail specified by that resolution range. This is shown on the right panel in Figure 3. At all resolution ranges, `CrysFormer` predictions are clearly better than that of the `U-Net`-based models. In particular, for `CrysFormer`, we still have a majority of predictions with phase error $< 60°$ even at the highest ranges of resolution.

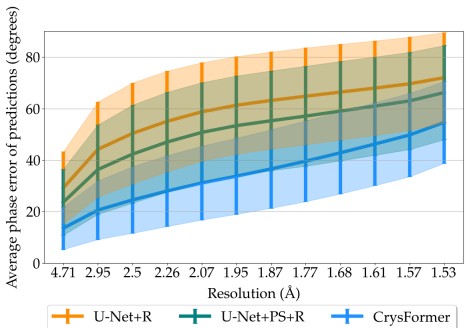 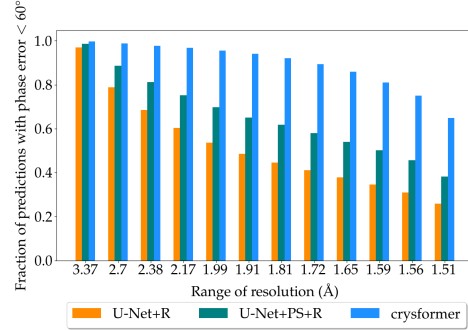

Figure 3: Dipeptide dataset. **Left**: Average phase error of model predictions against reflection resolution. **Right**: Fraction of model predictions for which phase error is $< 60°$ at various ranges of resolution.

**Results on 15-residues.** On our dataset of 15-residue examples, which consisted of only $165,858$ training and $16,230$ test cases (less than one-tenth the size of our dipeptide dataset), we trained for 80 epochs to a final average test set Pearson correlation of about $0.747$. We then performed a refining training run of 20 epochs, incorporating the original training run's predictions as additional input channels when training the `CrysFormer`, and obtained an improved average test set Pearson correlation of about $0.77$ and phase error of about $67.66$. On both of these runs, we used the Nyström approximate attention mechanism (Xiong et al., 2021) when incorporating our partial structure information to reduce time and space costs. Even still, each training epoch still took about $6.28$ hours to complete. Thus due to time considerations, we decided not to attempt to train a U-Net on this dataset for purposes of comparison.

We provide visualizations of some model predictions in Figure 4; more can again be found in the appendix. We also plot the average mean phase errors of the predictions of our models against reflection resolution, as well as the fraction of our models' predictions for which the calculated mean phase error is $< 60°$ at various ranges of resolution in Figure 5. These results show that this is a more difficult dataset with reduced sample size; yet `CrysFormer` predictions tend to accurately reproduce details of the desired electron densities.

We used the *Autobuild* program within the *PHENIX* suite (Terwilliger et al., 2008; Liebschner et al., 2019) to perform automated model

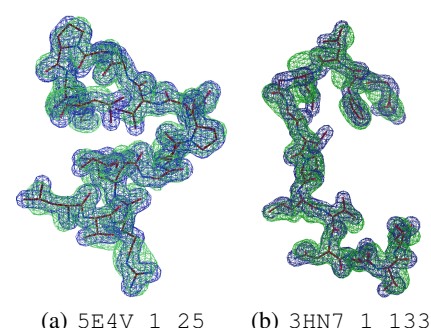

(a) `5E4V_1_25`    (b) `3HN7_1_133`

Figure 4: Visualization of two successful predictions after a refining training run; ground truth density maps shown in blue and predictions shown in green.

building and crystallographic refinement on a randomly selected subset of 302 test set predictions after the refining training run. We found that 281 out of 302 ($\sim 93\%$) refined to a final atomic model with a crystallographic $R$-factor of less than $0.38$, indicating success, when solvent flattening was applied. Without solvent flattening, 258 out of 302 ($\sim 85\%$) refined to such an $R$-factor (performing solvent flattening is known to be especially effective for unit cells with high solvent content, i.e. a large amount of empty space around the atoms). Figure 6 shows these results as scatterplots; clearly only a small fraction of the subset of predictions did not refine successfully. And even if no

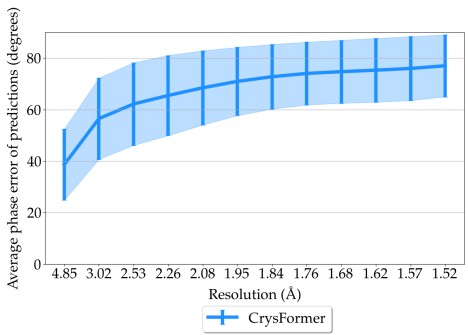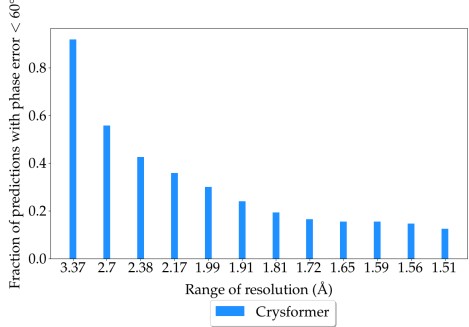

Figure 5: **Left**: Average phase error of model predictions on 15-residue dataset against reflection resolution. **Right**: Fraction of model predictions on 15-residue dataset for which phase error is $< 60°$ at various ranges of resolution.

refinement was performed at all, and instead an atomic model was repeatedly fit to our predicted electron densities, we found that 229 out of 302 ($\sim 76\%$) of the best such atomic models still had a crystallographic $R$-factor of less than $0.38$.

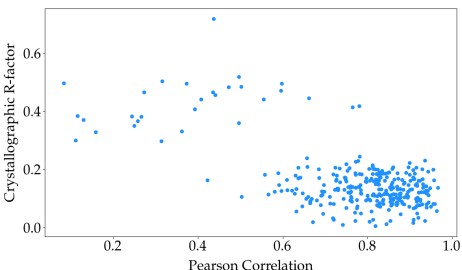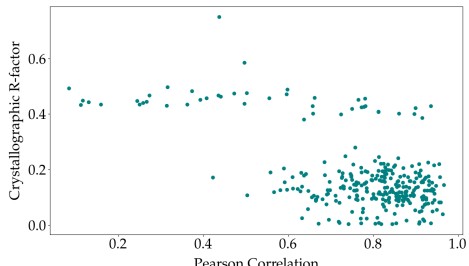

Figure 6: **Left Panel**: Scatterplot of post-refinement model R-factors, with solvent flattening applied. **Right Panel**: Scatterplot of post-refinement model $R$-factors, without solvent flattening applied

Furthermore, after automatic map interpretation using the autobuilding routines in *shelxe* (Usón & Sheldrick, 2018) to obtain a poly-alanine chain from each of the $16230$ test set predictions, we found that almost $74\%$ of the resulting models had calculated amplitudes with a Pearson correlation of at least $0.25$ to the true underlying data. Historical results indicate that further refinement would very likely produce a "correct" model if the initial poly-alanine model has at least such a correlation.

## 6 DISCUSSION

We have shown that `CrysFormer` outperforms state of the art models for predicting electron density maps from corresponding Patterson maps in all metrics on a newly introduced dataset (dipeptide). Overall, `CrysFormer` requires fewer epochs to reasonably converge and has a smaller computational footprint. Furthermore, our "refining" procedure greatly improves training for the vanilla `U-Net` architecture on our dipeptide dataset, as well as for training `CrysFormer` on our both dipeptide and 15-residues dataset.

**Limitations and next steps.** Following successful results on our initial 15-residue dataset, we also suggest training our model on variable unit cells at that problem size as future work. Eventually, we also prefer handling variable cell angles as well, moving beyond the orthorhombic crystal system. We will explore changing the formulation of our partial structures to have more than one amino acid residue in a structure, as having each partial structure representing only a single residue may no longer be reasonable, both computationally and from a practical perspective.

**Broader Impacts.** Solving the crystallographic phase problem for proteins would dramatically reduce the time and expense of determining a new protein structure, especially if there are no close homologs already in the Protein Data Bank. There exist some methods that sometimes work under special conditions Jiang et al. (2018), or that work sometimes but only at very low resolutions David & Subbiah (1994). The recent line of work on AlphaFold Jumper et al. (2021); Tunyasuvunakool et al. (2021) definitely helps in these problems; we note though that this is true mostly in cases where reliable predictions are possible due to strong homologs and/or extensive sequence data.

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
