# CRYSFORMER: PROTEIN STRUCTURE PREDICTION VIA 3D PATTERSON MAPS AND PARTIAL STRUCTURE ATTENTION

## A  MODEL ARCHITECTURE OF BASELINE U-NET MODEL

Our U-net architecture can be divided into three phases. The ***Encoding Phase*** consists of two 7x7x7 convolutional layers. The first has 25 output channels while the second has 30 output channels, and both are followed by the standard batch normalization and a ReLU activation. Then, a max pooling operation with kernel size 2x2x2 and stride 2 is used to reduce the height, width, and depth dimensions by a factor of 2.

The ***Learning Features Phase*** consists of a sequence of 7 residual blocks. Each of these blocks consists of a 7x7x7 convolutional layer with 30 output channels followed by batch normalization and ReLU activation, and then another 30-channel 7x7x7 convolutional layer with batch normalization but no activation. A squeeze and excitation block Hu et al. (2018) occurs at this point, applied with the channel dimension reduced by a factor of 2. Afterward, the residual skip connection is applied, followed by another ReLU activation. At the end of this phase, a naive upsampling operation is used to increase the height, width, and depth dimensions by a factor of 2, thus restoring the original dimensions.

The ***Decoding Phase*** consists of two 5x5x5 convolutional layers. The first has 25 output channels and is followed by batch normalization and a ReLU activation, while the second produces the model predictions and thus has only a single output channel. Since all elements of the target outputs were constrained to be in the range [-1, 1], we apply a final tanh activation function after this layer.

In all convolutional layers, the input is "same" padded to preserve height, width, and depth dimensionality. Also, the convolutional layers in the encoding and learning features phases are padded using torch's circular padding scheme to account for the periodic nature of the input Patterson maps. Furthermore, all convolutional layers were initialized using the kaiming_normal function of the default torch.nn module He et al. (2015). As with the `CrysFormer`, our U-net model is robust to training batches of examples with differing height, width, and depth shapes.

## B  ADDITIONAL DETAILS ON DATASET GENERATION

To start preparing our dataset, we selected nearly 24000 representative Protein Data Bank (PDB) entries using the following criteria: proteins solved by X-ray crystallography after 1995, sequence length $\geq 40$, refinement resolution $\leq 2.75$, refinement R-Free $\leq 0.28$, with clustering at 30% sequence identity. The standardized modifications we applied to each viable coordinate file were as follows: all temperature factors were set to 20, any selenomethionine residues were rebuilt as methionine, and all hydrogen atoms were removed leaving only carbon, nitrogen, oxygen, and potentially sulfur.

In our dataset generation process, an effort was taken to ensure diversity by sampling from PDB entities with low sequence similarity to each other. However, both test and training sets are taking random samples from the conformations allowed in rotamer and Ramachandran space. Any similar conformations would be expected to be in a different rotational orientation in the cell by the nature of the selection process. We did not compute all-versus-all clustering or force the test and training sets to sample distinct conformational regions. For our 15-residue dataset, in order to obtain a greater amount of starting coordinate files, we allowed at most 3 residues to be shared between distinct examples. To prevent potential overfitting that could arise from this sharing of subsegments, we

enforced that all examples derived from the same initial .pdb file would be placed together in either the training or test set.

Another issue regarding ambiguity in Patterson map interpretation is the fact that an electron density will always have the exact same Patterson map as its corresponding centrosymmetry-related electron density. Hurwitz (2020) provided a workaround that involved combining a set of atoms with the set of its centrosymmetry-related atoms into a single example output. However, this also requires a separate post-processing algorithm to separate the original and centrosymmetric densities for each of his model's predictions. Since we are working with real-world structures –rather than randomly placed data– we can exploit their known properties. In particular, we know that all proteinogenic amino acids are naturally found in only one possible enantiomeric configuration (Helmenstine, 2021). Although the mirror-image symmetry of enantiomers is not exactly the same as centrosymmetry, we show that this is enough to allow us to work with true electron densities of protein fragments.

## C  DESCRIPTION OF DATASET SUBSET

Due to limitations of online storage space, we provide a subset of our generated dataset. This subset represents a total of 200000 dipeptide examples. As expected, patterson.tar.gz contains the generated Patterson maps, while electron_density.tar.gz contains the corresponding electron densities. Meanwhile, partial_structure.tar.gz contains both of the partial structures for each dipeptide example in the subset.

The dataset can be downloaded through this link:

https://drive.google.com/drive/folders/1X7YkxDd7yTC1RTG1z3NbdRIfKLfFtkrx?usp=share_link

We will also provide a dataset of prepared .pdb coordinate files of 15-residue examples, to which our dataset generation process can be applied in order to produce Patterson map and electron density tensors.

## D  ADDITIONAL VISUALIZATIONS OF MODEL PREDICTIONS

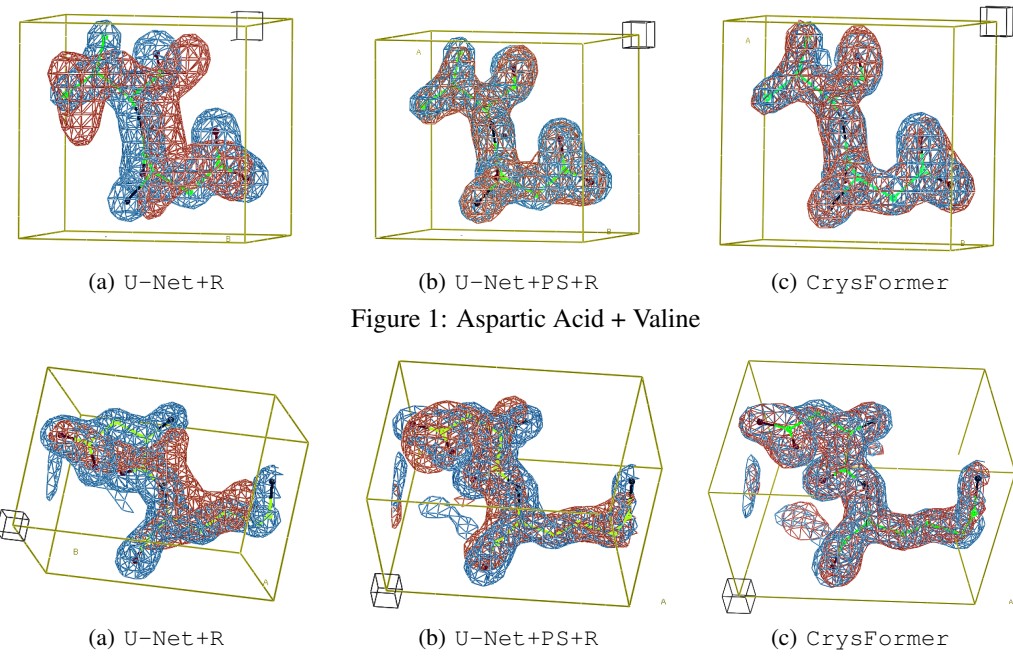

(a) U-Net+R    (b) U-Net+PS+R    (c) CrysFormer

Figure 1: Aspartic Acid + Valine

(a) U-Net+R    (b) U-Net+PS+R    (c) CrysFormer

Figure 2: Aspartic Acid + Lysine

Figure 3: Visualizations for dipeptide dataset. Ground truth density maps are shown in blue, while predictions are shown in red. The model used to generate the ground truth electron density is shown in stick representation for reference.

Figure 1 represents an example in which the additional partial structure input channels provided to the U-Net provided a substantial increase in prediction quality, allowing it to produce a prediction similar to that of the `CrysFormer`. Figure 2 represents an example in which both providing additional input channels to the U-Net and switching to `CrysFormer` provided noticeable improvements in prediction quality.

It is clear that as prediction quality increases as indicated by reported Pearson correlation, finer details of the true underlying structure are more likely to be accurately reproduced. The predictions in Figure 4 (e), (f), (g), and (h), as well as Figure **??** (a) [rank 55%] and (b) [rank 82%], were all successfully refined using all of the mentioned autotracing and refinement procedures. But even for relatively poor predictions such as (a) and (b), the rough overall shape can be reproduced even though several portions have clear inaccuracies.

Figure 5 shows the scatterplot of *shelxe* poly-alanine autotracing results on the full 15-residue test set. As mentioned, examples for which the amplitudes calculated from the initial poly-alanine chain built into the model electron density prediction have a Pearson correlation coefficient with the true underlying structure factor amplitudes of over 0.25 (shown above the red line) are extremely likely to be successfully refined.

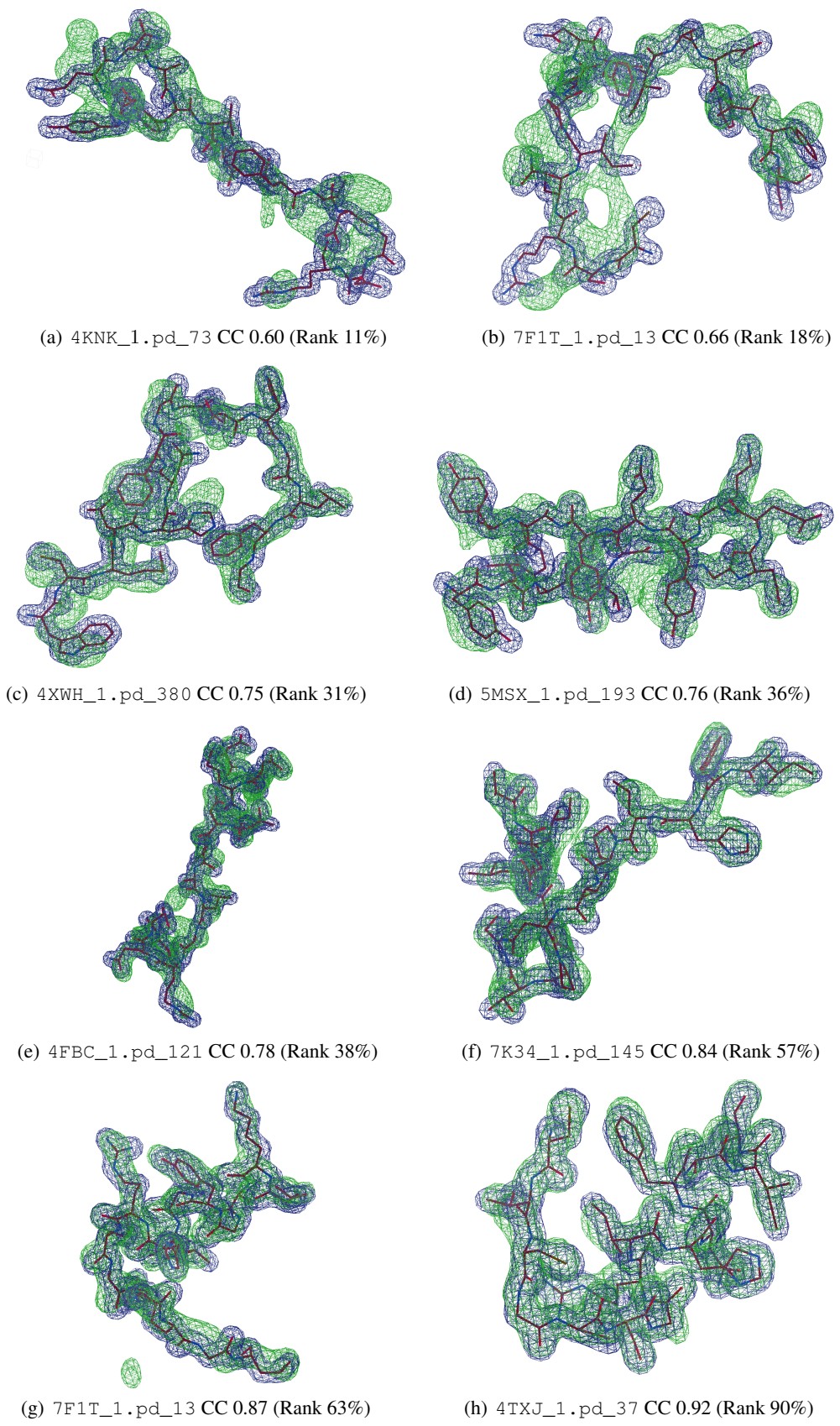

(a) `4KNK_1.pd_73` CC 0.60 (Rank 11%)

(b) `7F1T_1.pd_13` CC 0.66 (Rank 18%)

(c) `4XWH_1.pd_380` CC 0.75 (Rank 31%)

(d) `5MSX_1.pd_193` CC 0.76 (Rank 36%)

(e) `4FBC_1.pd_121` CC 0.78 (Rank 38%)

(f) `7K34_1.pd_145` CC 0.84 (Rank 57%)

(g) `7F1T_1.pd_13` CC 0.87 (Rank 63%)

(h) `4TXJ_1.pd_37` CC 0.92 (Rank 90%)

Figure 4: Visualizations for 15-residue dataset. Ground truth density maps are shown in blue, while predictions are shown in green. The model used to generate the ground truth electron density is shown in stick representation.

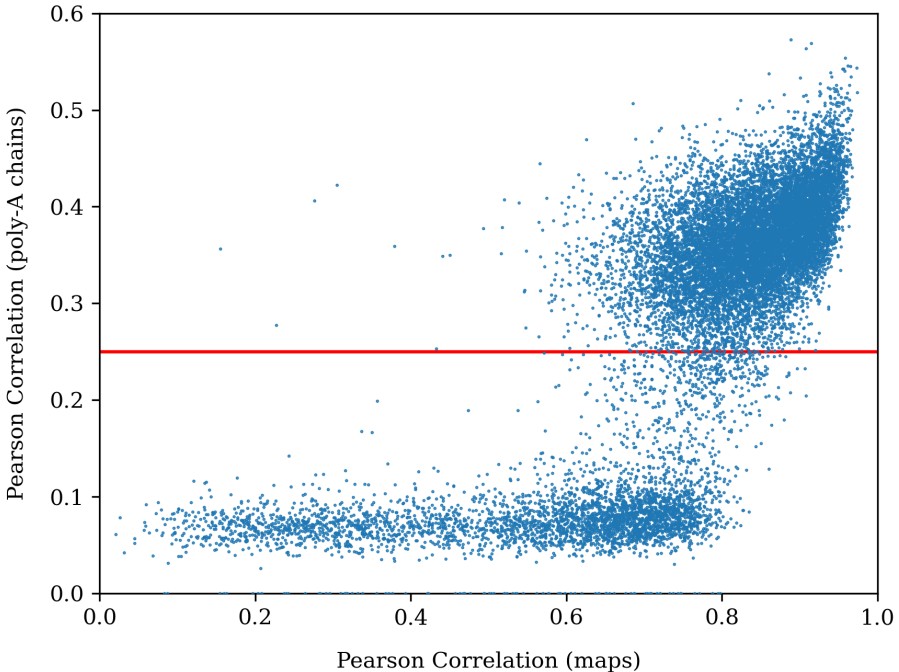

Figure 5: Scatterplot of the Pearson correlations of amplitudes of the poly-alanine chains autotraced by *shelxe* to the ground truth amplitudes vs the Pearson correlation of the predicted and ground truth maps for all 16,203 test cases