# OpenReview forum: "CrysFormer: Protein Structure Prediction via 3d Patterson Maps and Partial Structure Attention"
_ICLR.cc/2024/Conference — ICLR 2024 Conference Withdrawn Submission_

### Official Review · Reviewer_Rj5s · 2023-10-28

**Soundness:** 3 good
**Presentation:** 2 fair
**Contribution:** 3 good
**Rating:** 3
**Confidence:** 3

**Summary:**

Based on the developments of protein structure prediction, models can utilize a substantial dataset including sequence information and associated protein structures, but this paper proposes to the possible utilization of other accessible prior knowledge, such as protein crystallography and the incomplete structure of amino acids. Authors proposed CrysFormer, a novel transformer-based model that leverages protein crystallography and partial structural information for the purpose of predicting electron density maps of proteins. CrysFormer can infer electron density maps and a relative new dataset of variable-cell dipeptide fragments is built, consisting of peptide fragments with lengths of 2 and 15 residues.

**Strengths:**

1. This paper points out a new direction and problem, that the predicted protein structures are not yet complete, and X-ray crystallographic data can be used, which is a gold standard for critical details describing chemical interactions of proteins. CrysFormer can process the global information in Patterson maps to infer electron density maps.
2. It seems that CrysFormer achieves a significant improvement compared with baseline models, by incorporating “partial structure” information.
3. A new dataset about Patterson and electron density maps is built from PDB, which will be publicly available.

**Weaknesses:**

1. The predicted protein structures from computational methods without experimentally confirming data are incomplete and suffer from a lack of accuracy in many of the details. Could you explain why these predicted structures are incomplete? And what details are lacking? How can the proposed method, CrysFormer, solve these problems by incorporating X-ray crystallographic data? e.g., the predicted structures from CrysFormer have the details, but structures from other methods are ignored?

2. Why can the convolutional U-Net-based architectures in Pan et al. (2023) incorporate “partial structure” information, but CrysFormer architecture still leads to better reconstructions? Which part makes the improvements and differences?

3. what the \mathfrak{R} means? Why is this equation true for p? Could you provide mathematical demonstrations? How do you connect the Fourier transform and your model architecture? It seems that the Patterson function and the deep learning model are independent.

4. For Design Principle #2, when is the partial structure information available, and when it is not available? When the partial structure information is not available, could the predicted structures be used? How does the Crysformer in Figure 1 work under this condition?

5. When building the dataset, why dipeptides (two residues) and 15 residues are randomly selected? What are the criteria for this selection? From the successful results on the initial 15-residue dataset, why the results of the 15-residue dataset are better? Does this mean that more residues have better results?

6. The main results in Table 1 only include U-Net and the proposed models, which may be better to have more comparison methods, including the methods you have mentioned in the related work. From the title "Protein Structure Prediction via 3D Patterson Maps and Partial Structure Attention", it seems that this paper aims to do protein structure prediction, but there are no results about this.

7. How to evaluate the performance of the visualization results of density maps, shown in Figure 2 and Figure 4.

**Questions:**

See questions in the Weakness part.

---

### Official Review · Reviewer_S4uz · 2023-10-29

**Soundness:** 3 good
**Presentation:** 2 fair
**Contribution:** 3 good
**Rating:** 6
**Confidence:** 4

**Summary:**

The paper proposes a transformer-based model CrysFormer for accurate predictions of the electron density map of proteins with reduced computation costs by utilizing protein crystallography and partial structure information. CrysFormer processes the global information in Patterson maps and incorporates partial structure information by a tailored attention mechanism, resulting in significant improvement in model performance in electron density predictions. In addition, a novel dataset of variable-cell dipeptide fragments is introduced in the manuscript.

**Strengths:**

Originality: The paper uniquely contributes to the field by proposing a novel method for predicting the electron density using the global information in Patterson maps. In addition, the author curated a novel dataset, which could serve as a starting point of future research.

Quality: The paper carefully designs the experiments to support the idea and make clear visualizations.

Clarity: The paper effectively communicates its ideas and findings with clarity. The paper is well-written, and the logic is coherent. The core concepts and formulas are described in detail with clarity.

Significance: The model proposed in the integrate X-ray diffraction pattern for predicting electron density map, which is the first transformer-based model designed for such tasks. The extensive experimental results illustrate the effectiveness of the transformer encoder and the introduction of partial structures in the inputs, which could inspire further research in determining the protein structure.

**Weaknesses:**

1. The story itself is complete and coherent; however, it would be better if the authors can provide intuitions of the work: why is it useful to achieve high-accuracy predictions of protein electron map, and how will the model be used in broader applications, for example, protein design? In particular, the title and the abstract of the manuscript mention that the ultimate goal of the model is to predict protein structures, but there seems to be a gap between predicting the electron density and predicting the structure.

**Questions:**

1. The authors mentioned in the abstract that most methods for determining the structure of proteins rely on sequential information but tend to ignore structural information, which I also agree with. But I'm wondering why the authors chose the X-ray diffraction pattern as input: why not simply encode protein structure with atomic positions instead of inferring electron density, and if it's indeed necessary to predict electron density map, is it possible to use train a neural network which can achieve near-DFT accuracy?

2. The authors claimed that the tokens from the additional partial structure are from the initial embeddings and will not be passed to the following layers because they are used as a stable reference to attend in each layer. However, I doubt whether this really helps improve model performance: while the embeddings of Patterson Map are updated in the latent space, it seems more acceptable to me that the embeddings from partial structures should also be updated.

3. Can the authors clarify the dataset sizes? In the section New Datasets, about 24000 protein structures are picked out, while the dipeptide dataset seems to have 2M data in total. Besides, is there any reason that the datasets are split into training and test sets instead of training, validation, and test sets, or implementing k-fold cross-validation?

4. Why is there a lack of comparison with baselines for experiments on 15-residues?

5. How does the attention work when performing refining training? To be specific, how are the embeddings from the original training run's predictions concatenated to the input, and are the embeddings kept constant for stable reference or updated in each layer?

6. The authors provide abundant evidence to illustrate the reduced computation costs of CrysFormer in terms of the time needed, but I'm wondering how the memory cost of the model compared with baseline models.

---

### Official Review · Reviewer_nFiA · 2023-10-30

**Soundness:** 3 good
**Presentation:** 3 good
**Contribution:** 3 good
**Rating:** 3
**Confidence:** 3

**Summary:**

This paper focuses on enhancing protein structure prediction with the help of additional resources such as protein crystallography and partial structure of amino acids. The authors purport to be the first to deploy transformer-based models that weave in aspects of protein crystallography and partial structure data to derive electron density maps of proteins. They test their theories through experiments carried out on their unique datasets comprised of small peptide fragments.

**Strengths:**

- This research introduces a fresh dataset that could potentially illuminate how to integrate partial structure with protein crystallography in order to predict density maps.
- The CrysFormer shows potential in handling the global information of the Patterson map, and can also assimilate information from available partial structures.

**Weaknesses:**

- An ablation study examining the individual components of CrysFormer is missing. The effectiveness of using a partial structure should be evaluated in the context of the CrysFormer’s backbone. You have utilized a refinement strategy with the Unet structure, why not incorporate this into CrysFormer as well? Which is the major performance contributor? Protein crystallography, partial structure information, or sequence information?
- The lack of real cases in the dataset might considerably constrain the efficacy of your proposed method. Why haven't the authors utilized datasets of current crystal structures for small proteins? What hurdles have they come across in this regard? Are the obstructions due to a shortage of data, challenges in generalization, or issues with the model's scalability?
- The authors have mentioned AF2 in the abstract, but the content of the manuscript does not seem to have an apparent relation to AF2. In order to justify the additional information required, the CrysFormer, at the least, should significantly outperform AF2 and other folding methods. This particular point, however, was not confirmed by the authors. What about using AF2's result as an initial result?

**Questions:**

See weaknesses.

---

### Official Review · Reviewer_ATj6 · 2023-11-06

**Soundness:** 3 good
**Presentation:** 2 fair
**Contribution:** 2 fair
**Rating:** 3
**Confidence:** 3

**Summary:**

This work proposes CrysFormer, the first transformer-based model that utilizes protein crystallography and partial structure information to directly predict electron density maps of proteins. The model incorporates global information via self-attention mechanism and partial structure data. The authors introduce new datasets of small peptide fragments for evaluating electron density map prediction. Experiments show CrysFormer achieves higher accuracy predictions compared to convolutional baselines, while requiring less data and computations.

**Strengths:**

The use of a transformer Encoder-only design captures global dependencies well for this task compared to CNNs. The proposed self-attention mechanism is tailored for incorporating partial structure information and reduces computational costs. Another strength is the introduction of new datasets derived from PDB structures to benchmark performance, helping drive progress on this problem. The visualization of predictions provides intuition on where CrysFormer improves over baselines. The experiments are thorough, with quantitative metrics, convergence analysis and phase error analysis.

**Weaknesses:**

Although the results are promising when compared to the baselines, I feel the technical contrition related to deep learning of this work is limited. The proposed method is based on Vision transformer with a self-attention mechanism that can process Patterson maps and partial protein structures.

Furthermore, I have some reservations about the significance of the problem addressed in this work, as it is a newly defined problem with only one existing related study can be found.

Suggestion: In Figure 2, it would be beneficial to incorporate visualization results of the U-Net baseline. Additionally, for improved clarity, consider including numerical metrics alongside each visualization. This would be helpful since distinguishing the differences between Fig. 2d, e, and f is challenging when relying solely on visual inspection.

**Questions:**

Do the colors (yellow and blue) of the 3d patches shown in Fig. 1 have exact meanings?